# Mathematical Modeling of Within-Host, Untreated, Cytomegalovirus Infection Dynamics after Allogeneic Transplantation

**DOI:** 10.3390/v13112292

**Published:** 2021-11-16

**Authors:** Elizabeth R. Duke, Florencia A. T. Boshier, Michael Boeckh, Joshua T. Schiffer, E. Fabian Cardozo-Ojeda

**Affiliations:** 1Fred Hutchinson Cancer Research Center, Vaccine and Infectious Disease Division, Seattle, WA 98109, USA; f.boshier@ucl.ac.uk (F.A.T.B.); mboeckh@fredhutch.org (M.B.); jschiffe@fredhutch.org (J.T.S.); ecojeda@fredhutch.org (E.F.C.-O.); 2Department of Medicine, University of Washington, Seattle, WA 98195, USA; 3Department of Infection, Immunity and Inflammation, UCL Great Ormond Street Institute of Child Health, University College London, London WC1N 1EH, UK

**Keywords:** CMV kinetics, virus dynamics model, allogeneic transplantation

## Abstract

Cytomegalovirus (CMV) causes significant morbidity and mortality in recipients of allogeneic hematopoietic cell transplantation (HCT). Whereas insights gained from mathematical modeling of other chronic viral infections such as HIV, hepatitis C, and herpes simplex virus-2 have aided in optimizing therapy, previous CMV modeling has been hindered by a lack of comprehensive quantitative PCR viral load data from untreated episodes of viremia in HCT recipients. We performed quantitative CMV DNA PCR on stored, frozen serum samples from the placebo group of participants in a historic randomized controlled trial of ganciclovir for the early treatment of CMV infection in bone marrow transplant recipients. We developed four main ordinary differential Equation mathematical models and used model selection theory to choose between 38 competing versions of these models. Models were fit using a population, nonlinear, mixed-effects approach. We found that CMV kinetics from untreated HCT recipients are highly variable. The models that recapitulated the observed patterns most parsimoniously included explicit, dynamic immune cell compartments and did not include dynamic target cell compartments, consistent with the large number of tissue and cell types that CMV infects. In addition, in our best-fitting models, viral clearance was extremely slow, suggesting severe impairment of the immune response after HCT. Parameters from our best model correlated well with participants’ clinical risk factors and outcomes from the trial, further validating our model. Our models suggest that CMV dynamics in HCT recipients are determined by host immune response rather than target cell limitation in the absence of antiviral treatment.

## 1. Introduction

Cytomegalovirus (CMV) is a human herpes virus, HHV-5, that is transmitted through saliva or breast milk, trans-placentally, and during organ or hematopoietic cell transplantation. CMV infects more than 50% of the world’s population, and as with other herpes viruses, CMV infects its host for life in a latent form [1]. While largely asymptomatic in the general population, CMV causes serious disease in neonates and immunocompromised hosts, including recipients of allogeneic hematopoietic cell transplants (HCT) in whom CMV causes pneumonia, gastroenteritis, and retinitis [1,2].

Intra-host mathematical modeling of CMV and other viral infections has proven critical to understanding the dynamics of virus-host interactions and allows for the simulation of clinical trials to improve the development of antiviral therapies and vaccines [3,4,5]. However, prior mathematical modeling of CMV has been limited by a lack of availability of untreated viral load data. Ganciclovir was approved for the treatment of AIDS retinitis in 1989 before the widespread adoption of quantitative polymerase chain reaction (PCR) for measurement of CMV viral load [6,7,8]. Thus, serious CMV infections are generally treated with antivirals, precluding studies of untreated natural infection with quantitative PCR. Previous studies analyzing CMV kinetics calculated viral doubling-times and decay half-lives and associated high CMV viral load, viral load slope, and CMV doubling-times with poor outcomes [9,10,11]. Kepler et al. developed a theoretical model from in vitro and literature values for CMV parameters [12]; Rose and colleagues modeled varying viral load responses to ganciclovir [13]; Mayer and colleagues developed deterministic and stochastic models of infant infection fit to viral load data from frequent sampling of the oral mucosa [14,15].

A mathematical model of CMV viral loads measured in blood from untreated patients is needed to characterize the natural history of CMV accurately. Such a model would quantify the main mechanisms driving the dynamics of CMV in the HCT population after transplant. A data-validated model may allow us to understand how antiviral treatments reduce viral replication in this setting and then further simulate treatment and dosing scenarios to optimize viral suppression and to lower risk of disease after HCT. Here, we present a mathematical model fit to viral load data obtained from frozen serum samples from the placebo group from the only randomized controlled trial of ganciclovir for the early treatment of CMV in bone marrow transplant recipients [16]. Because these participants were not treated with ganciclovir until they reached the study endpoint of tissue-invasive CMV disease or death, we were able to capture the full dynamic range of CMV viral loads in our model. Following HCT, viral load trajectories and responses to antiviral treatments varied widely among transplant recipients likely due to great variability in immune parameters. Modeling infection in the absence of treatment allowed us to capture the natural variability in these immune parameters and associate their values with clinical outcomes from the trial.

## 2. Materials and Methods

### 2.1. Study Approach

We analyzed CMV viral loads from viral episodes that occurred in HCT recipients after transplant to characterize the natural history of CMV in untreated individuals. We developed four mechanistic ordinary differential Equation (ODE) mathematical models of within-host CMV infection each with specific underlying mechanistic assumptions regarding the dynamics of CMV-susceptible cells and CMV-specific immune responses during infection. We used model selection theory to compare multiple instances of these models. Specifically, in the model selection process, within each main model, we chose which parameters could plausibly be zero in a biological setting and set those to zero individually and in combination such that we fit every combination of biologically feasible parameters to determine what version of the model the data supported most strongly. We obtained two parsimonious models with identifiable parameters from the competing list to describe the data. We then used the model with the most biological plausibility to validate the model parameters by assessing their associations with risk factors for CMV infection and disease and the primary endpoint of the clinical trial—the development of CMV disease by 100 days after HCT.

### 2.2. Clinical Data

Frozen serum samples were saved from participants in the placebo-controlled randomized controlled trial of ganciclovir for early treatment of CMV after HCT [16,17]. In this trial, viral cultures were used to screen for CMV in allogeneic HCT recipients who were either CMV seropositive or who had received marrow from CMV seropositive donors. If viral cultures were positive prior to day 80 after transplant and tissue-invasive CMV disease had not already by diagnosed, HCT recipients were randomized to receive ganciclovir or placebo through day 100 post-transplantation. Participants were followed for primary and secondary outcomes of tissue-invasive CMV disease or death, respectively [16]. In a recent follow-up study to this historic trial, Duke and colleagues obtained these frozen samples from the Fred Hutch Infectious Disease Sciences Biospecimen Repository and selected samples for testing at approximately weekly intervals from day 0 to 100 after transplantation. The University of Washington Molecular Virology Laboratory performed quantitative CMV DNA PCR testing using a laboratory-developed assay with limit of quantification of 71.4 IU/mL and limit of detection of 35.7 IU/mL [17].

Here, we analyzed viral loads obtained from participants in the placebo group prior to diagnosis of CMV disease (at which point participants received treatment with open-label ganciclovir). For each individual, we did not include undetectable viral loads measured before the first positive aside from the last negative observation. If there was more than one viral episode in an individual with negative samples between episodes, we analyzed only the episode with the larger amount of data points. We did not model viral load data from participants who had only undetectable viral loads (n = 2).

### 2.3. Calculation of Viral Load Kinetics

Peak viral load was considered to be the maximum log-10 converted viral load measured during the viral episode. For those participants who had undetectable CMV viral loads after HCT, we included only the last undetectable observation prior to the first positive viral load measured. Because the time when the CMV virus first became detectable between these measurements is unknown, we considered the start of the viral episode to be the midpoint between the last undetectable and first positive viral load. For participants who cleared the virus (i.e., viral load returned to undetectable), we considered the end of the episode to be the midpoint between the last positive and subsequent undetectable viral load. The expansion slope to the first positive was calculated as the difference in log-10 converted viral loads (i.e., viral load value at first positive minus the limit of detection) divided by the difference in times when the first positive viral load was measured and the start of the episode. Likewise, the expansion slope to the peak was calculated as the difference in the peak and the limit of detection divided by the difference in times between when the peak viral load was measured and the start of the episode. The clearance slope was the difference in the limit of detection and the peak viral load divided by the difference in times between when the episode ended and when the peak viral load was measured.

### 2.4. Model with Target Cell Limitation and Implicit, Static Immune Control (TC, No EIS)

The first of four main ODE models we used to understand the natural history of untreated CMV during HCT was the standard within-host model of virus dynamics (Figure 1a) [3]. This model includes three main compartments: Cells susceptible to CMV (S), CMV-infected cells (I), and CMV virions (V). Susceptible cells (S) expand with constant rate λ, die at rate $\delta_{S}$, and are infected by CMV with rate β. CMV-infected cells (I) die at rate $\delta_{I}$, assumed implicitly to include a static immune response against CMV. Finally, CMV-infected cells (I) produce at rate π virions (V) that are cleared at rate γ. Under these assumptions, the model has the form
(1)$$\begin{array}{c} \frac{dS}{dt}=\lambda -\delta_{S}S-\beta VS\\ \frac{dI}{dt}=\beta VS-\delta_{I}I\;\\ \frac{dV}{dt}=\pi I-\gamma V \end{array}$$

### 2.5. Model with Target Cell Limitation and an Explicit, Dynamic Immune System (TC, EIS)

The second main model (*TC*, *EIS*) adds an explicit immune response that changes over time (explicit immune system = *EIS*) to the target cell-limited model in Equation (1). The model schematic is shown in Figure 1b. We assumed there is a CMV-specific effector cell compartment (E) that expands in the presence of CMV virions (V) at rate ω and decays with rate $\delta_{E}$ [18,19,20,21,22]. Effector cells (E) kill CMV-infected cells (I) at rate κ. Because this model includes an explicit, cytotoxic immune response against CMV, the parameter $\delta_{I}$ in this model may represent an innate response to infection or an intrinsic death rate of the infected cells (I) due to viral infection or both. With these assumptions, the *TC*, *EIS* model is:(2)$$\begin{array}{c} \frac{dS}{dt}=\lambda -\delta_{S}S-\beta VS\\ \frac{dI}{dt}=\beta VS-\delta_{I}I-\kappa IE\;\\ \frac{dV}{dt}=\pi I-\gamma V\\ \frac{dE}{dt}=\omega V-\delta_{E}E \end{array}$$

### 2.6. Explicit, Dynamic Immune Control Model without Target Cell Limitation (EIS, No TC)

The third main model (*EIS*, *no TC*) differs from the second in that we assume the susceptible cell pool (S) is so large that it is not changed significantly during CMV infection. This notion is plausible biologically given the large number of cell types that CMV infects and the high prevalence of those cell types in the human body [1]. Thus, we assumed in this model that the size of the susceptible cell compartment remains constant with concentration $S_{0}$, allowing us to remove the *S* Equation entirely. We simplify further by defining the composite parameter $\beta^{*}=\beta S_{0}$. Based on this assumption, the *EIS*, *no TC* model has the form:(3)$$\begin{array}{c} \frac{dI}{dt}=\beta^{*}V-\delta_{I}I-\kappa IE\;\\ \frac{dV}{dt}=\pi I-\gamma V\\ \frac{dE}{dt}=\omega V-\delta_{E}E \end{array}$$

The model schematic is shown in Figure 1c. For parameter identifiability purposes, we rescaled variables and parameters to consider three more related models. First, we rescaled the variable $\hat{I}=\pi I$ and introduced the composite parameter $\hat{\beta }=\pi \beta^{*}$ into Equation (3) so that the model takes the form:(4)$$\begin{array}{c} \frac{d\hat{I}}{dt}=\hat{\beta }V-\delta_{I}\hat{I}-\kappa \hat{I}E\;\\ \frac{dV}{dt}=\hat{I}-\gamma V\\ \frac{dE}{dt}=\omega V-\delta_{E}E \end{array}$$

Next, for further simplification, we considered the additional scaling $\hat{E}=\kappa E$ and introduced the composite parameter $\hat{\omega }=\kappa \omega$. Incorporating these definitions into Equation (4), the model becomes:(5)$$\begin{array}{c} \frac{d\hat{I}}{dt}=\hat{\beta }V-\delta_{I}\hat{I}-\hat{I}\hat{E}\;\\ \frac{dV}{dt}=\hat{I}-\gamma V\\ \frac{d\hat{E}}{dt}=\hat{\omega }V-\delta_{E}\hat{E} \end{array}\;$$

Alternatively, we considered the scaling $\hat{E}=\frac{E}{\omega }$ and $\hat{\kappa }=\kappa \omega$ in Equation (4), which results in the following model:(6)$$\begin{array}{c} \frac{d\hat{I}}{dt}=\hat{\beta }V-\delta_{I}\hat{I}-\hat{\kappa }\hat{I}\hat{E}\;\\ \frac{dV}{dt}=\hat{I}-\gamma V\\ \frac{d\hat{E}}{dt}=V-\delta_{E}\hat{E} \end{array}$$

### 2.7. Semi-Mechanistic, Explicit Immune Control Model (VE)

Finally, because of the possibility of overfitting the above models due to the large number of parameters, we constructed a fourth main model, a semi-mechanistic model for CMV viral and immune dynamics, by assuming that during CMV infection, viral dynamics are in quasi-stationary state with respect to the infected cell compartment (Figure 1d). Under this assumption, πI≈γV. We simplified the model in Equation (3) by combining the remaining viral and infected cell terms into one parameter: $r_{v}=\frac{\beta^{*}\pi }{\gamma }-\delta_{I}$. We called $r_{v}$ the CMV turnover rate. Under these assumptions, the model is:(7)$$\begin{array}{c} \frac{dV}{dt}=r_{v}V-\kappa EV\;\\ \frac{dE}{dt}=\omega V-\delta_{E}E \end{array}$$

To find an identifiable model that fit the data well, we further considered the rescaling $\hat{E}=\frac{E}{\omega }$ and the composite parameter $\hat{\kappa }=\kappa \omega$, resulting in the model:(8)$$\begin{array}{c} \frac{dV}{dt}=r_{v}V-\hat{\kappa }\hat{E}V\;\\ \frac{d\hat{E}}{dt}=V-\delta_{E}\hat{E} \end{array}$$

Alternatively, with the same goal of identifiability, we introduced the rescaling $\hat{E}=\kappa E$ and the composite parameter $\hat{\omega }=\kappa \omega$. Incorporating these assumptions into Equation (7) resulted in the model:(9)$$\begin{array}{c} \frac{dV}{dt}=r_{v}V-\hat{E}V\;\\ \frac{d\hat{E}}{dt}=\hat{\omega }V-\delta_{E}\hat{E} \end{array}$$

### 2.8. Population, Nonlinear, Mixed-Effects Approach

To fit the models to the CMV viral load observations, we used a nonlinear, mixed-effects framework. Under this framework, a viral load observation for individual i at time k is modeled as $\log_{10}y_{ik}=f_{V}\left(t_{ik},\theta_{i}\right)+\epsilon_{V}$. Here, $f_{V}$ represents the solution of the mechanistic model for the variable describing the virus (V) where $\theta_{i}$ is the parameter vector for individual i and $\epsilon_{V}\sim N\left(0,\sigma_{v}^{2}\right)$ is the measurement error for the log10-transformed viral load. We assumed that $\theta_{i}$ is drawn from a probability distribution with median or fixed effects $\theta^{pop}$ and random effects $\eta_{i}\sim N\left(0,\sigma_{\theta }\right)$. Unless otherwise specified, we modeled parameters as $\theta_{i}=\theta^{pop}e^{n_{i}}$. In other words, the modeled parameters are log-normally distributed among the population with variability denoted by $\eta_{i}$ such that $ln\left(\theta_{i}\right)=ln\left(\theta^{pop}\right)+\eta_{i}$.

We modeled the initial value for the variable V as $V_{i,0}=V_{0}^{pop}e^{\eta_{i}+\chi_{viremic}}$ for participants who were viremic (i.e., CMV viral loads were detectable) at the start of the modeling interval, which in this data set meant that they were viremic on first measurement after transplant. We estimated $\chi_{viremic}$ as a covariate of $V_{0}$, meaning that $\chi_{viremic}=0$ for the group of participants who were not viremic at the time of transplant (and thus at the time of the start of the modeled viremic episode), but that $\chi_{viremic}$ could be estimated as greater than zero for the group of participants who were viremic at the time of transplant (and at the start of the modeled viremic episode). Including $\chi_{viremic}$ as a covariate of $V_{0}$ allowed the population distribution for $V_{0}$ to be bimodal.

For viral load observations below the limit of detection we used the probabilistic model that Monolix software (www.lixoft.com accessed on 27 October 2021) provides for left-censored data [23].

### 2.9. Model Fitting

We explored the ODE models as described above by fitting versions of each model assuming some parameters were equal to zero and estimating the remaining ones, including initial conditions of state variables, as shown in Table A1, Table A2, Table A5 and Table A7. Certain parameters were estimated for each model and were never fixed at a value of zero because their values must be non-zero in order to sustain infection (β, π) or an immune response to infection (ω, κ). However, in the time frame modeled, susceptible cells may or may not proliferate (λ) or die ($\delta_{S}$), and infected cells ($\delta_{I}$) or effector immune cells ($\delta_{E}$) may or may not die at significant rates. Thus, we assign these parameters values of zero individually and in combination such that we fit all combinations of these parameters for each of the four main models that include these parameters. Thus, we explored a total of 38 individual models. For each model, we obtained the Maximum Likelihood Estimation (MLE) of the measurement error standard deviation $\sigma_{v}$, the MLE of the vector of fixed effects $\theta^{pop}$, and the MLE of the vector of standard deviations of the random effects $\sigma_{\theta }\;$ for each parameter using the Stochastic Approximation of the Expectation Maximization (SAEM) algorithm embedded in the Monolix software. We ran the SAEM algorithm five times (i.e., assessments) for each model using randomly selected initial values for the estimated parameters. For all model fits we assumed $t_{i0}=0$ as the time of last negative viral load after HCT or for those whose first viral load after transplant was positive, $t_{i0}=0$ coincided with the first viral load measured.

### 2.10. Model Selection

To determine the most parsimonious model, we calculated the log-likelihood (logL) for all five assessments for each of the 38 models. Then, we computed the Akaike Information Criterion (AIC) for the assessment with the highest logL, where $AIC=-2\max\left(\log L\right)+2m$ with m being the number of parameters estimated. Then, we defined the delta score $\Delta AIC_{j}=AIC_{j}-min\left(AIC\right)$, where $AIC_{j}$ is the particular AIC for a model, and $\min\left(AIC\right)$ is the minimum AIC from all the models compared. We assumed two models had similar support from the data if the delta scores comparing them was less than two, i.e., ΔAIC<2.

We analyzed the selection of our models further by assessing the relative standard error of the parameters for practical identifiability. During the estimation process, if a large change in a parameter causes no change in the likelihood, the data is not informing the value of the parameter under that specific structural model. The relative standard error (RSE) is a summary measure obtained during the parameter estimation process for each parameter and is small if the data provides adequate information to estimate the parameter. Specifically, Monolix calculates the Fisher Information Matrix (FIM) for each set of estimated parameters. In this matrix, the contribution of each parameter to the likelihood is indicated along the diagonal. The standard error (SE) vector is the square root of the diagonal of the inverse of the FIM. In that sense, the smaller the SE for a parameter, the more the data is informing that parameter. Finally, the RSE is the SE divided by the estimated parameter value such that the RSE is the uncertainty in estimation of a parameter normalized by its estimated value. When the SE of a parameter is greater than its estimated value (RSE > 100%), that parameter is generally regarded not to be practically identifiable [24]. To be stringent, for those models with parameters with RSE percentages above 50%, we attempted to reduce the number of parameters while still maintaining some biological plausibility to which we could map the parameter values. We chose final models based on AIC, but also on identifiability and biological plausibility.

## 3. Results

### 3.1. CMV Viral Load Kinetics and Modeling Strategy

To characterize the natural history of CMV in untreated individuals we analyzed CMV viral load observations from the stored serum of individuals in the placebo group of the randomized controlled trial of ganciclovir described above. Viral loads from the 35 HCT recipients in the placebo group are shown in Figure 2. For most participants, these viral loads were measured in the absence of antiviral treatment. For those HCT recipients who reached the primary clinical trial endpoint of tissue-invasive CMV disease in the first 100 days after HCT, open-label ganciclovir was offered. On-treatment and post-treatment viral loads are shown for those participants with dashed lines in Figure 2. In addition, two of the participants in the placebo group had only undetectable viral loads. We did not include their viral load data for modeling but modeled the viral load data of the remaining 33 HCT recipients. On average, 7.2 viral loads were measured during the modeling interval per HCT recipient (median 7), with the number of measurements ranging from 3 to 14 per recipient.

In the original trial, fifteen participants in the placebo group developed CMV disease by day 100 [16]. Two of these participants died so quickly of CMV disease that no additional points were able to be measured prior to their deaths (IDs 11, 30). Two participants were diagnosed with CMV disease just prior to day 100 (ID 56, 61), such that the full viral episode was able to be included prior to the start of open-label ganciclovir for ID 61; ID 56 had multiple episodes of viremia, so the first episode was removed independent of CMV disease. We did not model data after the start of open-label ganciclovir for eleven participants. However, two of these died shortly after diagnosis such that few points were omitted (ID 10: 2 points, ID 58: 3 points). For the nine participants who survived beyond day 100 (IDs 1, 5, 9, 17, 26, 29, 38, 60, 62), between 2 and 5 points were removed for an average of 3.7 points removed. For those participants with more than one viral episode with negative samples between episodes (IDs 3, 56, 63), we analyzed only the episode with the larger number of data points.

CMV viral load kinetics were calculated during the first five weeks of study treatment (ganciclovir or placebo), and their associations with CMV disease and death were examined in detail in a prior publication [17]. The focus of the current manuscript is to develop a within-host, mechanistic mathematical model of untreated CMV infection after HCT rather than a kinetic analysis or an analysis of clinical outcomes. However, we have characterized some basic kinetics of the modeled episodes to demonstrate the heterogeneity that a model would need to capture and for comparison to the literature. However, the existing literature is based on assays performed in whole blood prior to the development of the international standardization of PCR measurements (genomes or copies/mL rather than International Units/mL), whereas our testing was performed using a plasma assay on serum samples and were converted to IU/mL [9,11,25].

Generally, CMV viral load kinetics in the HCT recipients were heterogeneous in the absence of antiviral treatment (Figure 2), but we were able to classify them into four general categories: (1) rapid growth only, (2) rapid growth initially that later slowed, (3) growth followed by partial clearance during the observation period or (4) growth followed by complete clearance (Figure 3). We also observed that in categories (3) and (4), there was a plateau phase before viral clearance in some individuals. Of the 33 modeled episodes, seven exhibited rapid growth; four slowed growth; 12 partial clearance; and despite the profound immunosuppression required for bone marrow transplant at this time, ten were able to clear viral particles from the blood completely in the first 100 days after HCT. Granted, some of the growth categories are potentially misleading, as some participants developed CMV disease and were treated with ganciclovir and thus may have cleared virus via immune mechanisms had they not been treated with antivirals. However, given the high death rate of CMV disease without treatment, we suspect this is unlikely in most cases.

Figure 4 depicts the viral kinetics of the modeled episodes. Peak viral load ranged from $10^{2}$ to $10^{7.9}$ IU/mL with median $10^{4.5}$ IU/mL and IQR $10^{3.3}$ to $10^{5.7}$ IU/mL. The duration of modeled episodes ranged from 4.5 to 75 days with a median of 29 days and IQR 23.5 to 46.5 days. Note that duration is shown in weeks in Figure 4a. When the expansion slope was calculated from the beginning of the viral episode (see methods for calculation of start of episode) to the peak viral load (Slope to Peak), slope ranged from $0.06\;\log_{10}$ to $0.38\;\log_{10}$ IU/mL per day with median slope $0.17\;\log_{10}$ IU/mL per day (equivalent to a doubling time of 1.8 days) and IQR $0.10\;\log_{10}$ to $0.19\;\log_{10}$ IU/mL per day (Figure 4b). When the expansion slope was calculated from the beginning of the viral episode to the first positive viral load (Slope to First Positive), slope ranged from $0.01\;\log_{10}$ to $1.92\;\log_{10}$ IU/mL per day with median slope $0.22\;\log_{10}$ IU/mL per day (equivalent to a doubling time of 1.4 days) and IQR $0.13\;\log_{10}$ to $0.38\;\log_{10}$ IU/mL per day (Figure 4b). In the ten participants whose CMV viral loads returned to undetectable levels, the clearance slope was calculated from the peak viral load to the end of the viral episode (see methods for calculation of the end of episode) [26]. The clearance slope ranged from $-1.11\;\log_{10}$ to $-0.02\;\log_{10}$ IU/mL per day with median slope $-0.12\;\log_{10}$ IU/mL per day and IQR $-0.16\;\log_{10}$ to $-0.09\;\log_{10}$ IU/mL per day (Figure 4b).

Comparing our results to literature values, Emery and colleagues found peak viral loads to be somewhat lower, ranging from $10^{2.7}$ to $10^{6.0}$ genomes/mL with median $10^{3.9}$ genomes/mL among bone marrow transplant (BMT) recipients [11]. Emery et al. calculated the slope between the last negative and first positive viral load among all patients studied (a combination of renal, liver, and bone marrow transplant patients) and found that it ranged from $0.03\;\log_{10}$ to $1.65\;\log_{10}$ genomes/mL per day with median $0.24\;\log_{10}$ genomes/mL per day similar to the rate we calculated for the slope to first positive [11]. Emery and colleagues also reported that the half-life of decline of viral DNA in the blood of eleven BMT recipients was 1.52±0.67 days when receiving ganciclovir [9]. Interestingly, in those 10 participants who cleared virus spontaneously in our data set, the median viral decline of $-0.12\;\log_{10}$ IU/mL per day corresponds to a half-life of 2.51 days, slower than that calculated on ganciclovir.

To identify the possible mechanisms that drive the observed heterogeneous CMV kinetics, we developed and used model selection theory to rank four competing ODE models as described in the methods.

### 3.2. A Dynamic Effector Compartment Is Needed to Explain CMV Kinetics during HCT

To characterize the viral dynamics of CMV, initially, we used the following two mathematical models (Figure 1a,b): (1) the standard model of virus dynamics with target cell limitation but without an explicit effector immune system [3] (Equation (1), model numbers 1.1–1.8—*TC*, *No EIS* model), and (2) an adaptation of that model including an explicit immune cell compartment (Equation (2), model numbers 2.1–2.8—*EIS*, *TC* model). We used a nonlinear, mixed-effects approach to fit the two models under different assumptions regarding their parameter values (i.e., some parameters were given fixed values = 0) for a total of 16 models (models 1.1–1.8 and 2.1–2.8, Table A1 and Table A2). Then, we used model selection theory to identify the most parsimonious model from this set of competing models (see Materials and Methods, Section 2, for details). For target cell-limited models without *EIS*, lower AICs were obtained when the death rate for infected cells $\delta_{I}>0$ was estimated (models 1.5–1.8), implying the need for inclusion of static immune control or cytopathic cell death or both when dynamic immune control is not present. However, when an explicit, dynamic immune system was included in the *TC*, *EIS* models, the best models (red triangles in Figure 5; models 2.1 and 2.2 in Table A2) did not require $\delta_{I}$ to explain the data. Furthermore, from the different instances of the two models, we found that models with explicit immune responses against CMV explained the data more accurately and parsimoniously than models without one (Figure 5).

Model fits for the best model with and without an effector cell compartment are shown in Figure 6 and Figure A1, respectively, using individual parameter estimates in Table A3 and Table A4, respectively. Model fits for the best model without an effector compartment (model 1.8) show that this model lacks the necessary complexity to recapitulate the heterogeneity in the CMV dynamics (Figure A1) compared to the fits of the model including immunity (Figure 6).

In summary, we found that a model with an explicit, dynamic immune response against infection is required to recapitulate the heterogeneity of patterns observed in CMV kinetics after HCT.

### 3.3. Target Cell Limitation Is Not a Significant Driver of CMV Kinetics during HCT

Next, we explored whether having a dynamic, susceptible, target cell compartment was also necessary to explain the viral load data. We adapted the best, main model (Equation (2)) from the previous section and replaced the dynamic susceptible cell compartment with a static compartment. Because CMV can infect a large and diverse population of human cells (epithelial, endothelial, fibroblast, and smooth muscle cells, among others), in this model, we assumed that the number of susceptible cells is large and does not change significantly during infection [1]. Therefore, the susceptible cell compartment in this model is constant ($S\left(t\right)=S_{0}$ for all t) (Equation (3) and Figure 1c).

We explored several versions of the models in Equations (3)–(6); models 3.1–3.4, 4.1–4.2, 5.1–5.2, 6.1–6.7 (Table A5) and found that the best models with a constant number of susceptible cells (blue boxplots in Figure 7) generally outperform the models with dynamic susceptible cell compartments (orange boxplots in Figure 7). This result suggests that target cell limitation may not be a main driver of CMV dynamics during HCT.

Up until this point in the model selection process, we focused on comparing models to identify which mechanisms were driving the observed CMV dynamics. We determined that an explicit, dynamic, immune cell compartment is needed, but a dynamic target cell compartment may not be needed to recapitulate the data. However, next, we shifted our focus to parameter estimation and identifiability. When fitting to only one type of data, in this case, viral load, estimating distinct parameters that have similar effects on model control (e.g., the viral infectivity rate and viral production rate both contribute to viral expansion) may prove impossible. The data does not provide enough information to distinguish the contribution of separate parameters. In this case, some of the involved parameters might not be identifiable, and multiple values with similar goodness of fit can be estimated. To avoid this problem, parameters can be combined into composites that describe multiple rates simultaneously. The composite estimated values are identifiable, but measurable values for those parameters in the composites cannot be disentangled. We take this approach in Equations (4)–(6) and arrive at the model with the best AIC, model 6.2. Note that this model had the lowest AIC prior to the discovery that it contained non-identifiable parameters as described in the following paragraph.

Model 6.2 is based on Equation (6), which as with all models in this category, has a large and static target cell compartment and a dynamic effector cell compartment that expands in response to virus and kills infected cells. Specifically, Equation (6) combines the elements of viral expansion: infectivity, viral production rate, and the large target cell compartment into a single composite parameter $\hat{\beta }$ and the elements of effector cell killing: proliferation in response to virus and the killing rate into a single, composite parameter $\hat{\kappa }$. In this particular version of the model (6.2), $\delta_{E}$ = 0 and $\delta_{I}=0$, such that in this model we estimated only $\hat{\beta }$, γ, $\hat{\kappa }$, $I_{0}$, ${\hat{E}}_{0}$, $V_{0}$, and $\chi_{viremic}$ as shown in Equation (10). Thus, viral control is mediated by a combination of viral clearance γ and effector cell expansion and killing of infected cells $\hat{\kappa }$. As in all models we fit, because some participants had detectable viral loads on the first measurement after transplant, we estimated $V_{0}$ separately for those with and without initial detectable viral loads with $\chi_{viremic}$ as a covariate of $V_{0}$ (see Materials and Methods, Section 2, for details).
(10)$$\begin{array}{c} \frac{d\hat{I}}{dt}=\hat{\beta }V-\hat{\kappa }\hat{I}\hat{E}\;\\ \frac{dV}{dt}=\hat{I}-\gamma V\\ \frac{d\hat{E}}{dt}=V \end{array}$$

We examined the relative standard error % (RSE%) of this model as a further measure of identifiability and determined that some of the parameters still were not identifiable, as their RSE% were well above 50% (see Figure 8 for RSE% of model version 6.2) [24]. However, in fitting this version of the model, we found that estimates of ${\hat{E}}_{0}$ and $V_{0}$ were consistently at or near zero. Thus, we reduced the number of estimated parameters further in this model by fixing the values of the initial conditions of $\hat{E}$ and V to achieve model identifiability (Table A5). We found that fitting the version of the model with ${\hat{E}}_{0}$ and $V_{0}$ fixed at zero (model 6.7) resulted in a practically identifiable model with the lowest AIC, lower than the previously identified best model 6.2 (red triangles in Figure 7 and parameters on the right side of Figure 8).

The assumption that the concentrations of ${\hat{E}}_{0}$ and $V_{0}$ are near zero immediately prior to the start of viremia may be biologically plausible. Because we included the covariate $\chi_{viremic}$, all model $V_{0}$ values fixed at zero represent observed values below the limit of detection. In addition,$\;{\hat{I}}_{0}$ is small, but non-zero, and may represent the nidus of infection prior to viral production. HCT recipients with CMV viremia must rely either on CMV-specific T cell expansion from the small, remaining immune memory after conditioning or new, developing immunity from their donor after transplant, meaning that CMV-specific immunity may be weak after HCT. Indeed, Tormo and colleagues found that CMV-specific T cell responses were undetectable at the start of viremia in 13 out of 14 HCT recipients they studied, consistent with our assumption that ${\hat{E}}_{0}=0$ [18].

Model fits for the best model (6.7) are shown in Figure 9 using individual parameter estimates from Table A6. From the best model fits, we estimated that CMV has a median viral clearance rate in plasma of 0.27 virions per day, equivalent to a half-life of 2.6 days. Note that the viral clearance rate is mechanistically distinct from the slope of the viral decline as measured in the viral kinetics section, but the median estimate is similar to our kinetics calculation of viral decline of 2.5 days. Additionally, we found that in the best model, the death rate of the CMV-specific effector cells, $\delta_{E}$, was fixed at zero, suggesting long-lived immune cells.

### 3.4. A Semi-Mechanistic Model of CMV Virus and Immunity after HCT Is Identifiable and Offers Biological Plausibility

Because quantitative PCR viral load data from untreated participants with CMV infection is scarce and because CMV infection dynamics behave differently in different hosts (e.g., people with AIDS versus transplant recipients), parameter values have not been estimated robustly in the literature [10]. As noted in the previous section, fitting complete models with enough parameters to describe the data makes finding identifiable models with meaningful parameters difficult. This is problematic because when parameters are non-identifiable, we cannot rely on their estimates. In addition, despite having the lowest AIC, we noticed that the best model (6.7), was not capturing viral loads in those HCT recipients who did not clear virus and instead had persistently positive or increasing viral loads because this model does not have a non-zero viral steady state. Because these behaviors occur frequently in this patient population and will be important to capture clinically in simulation, we wanted to develop an identifiable model that could recapitulate this behavior.

Thus, we explored a semi-mechanistic mathematical model based on the previous model with an explicit and dynamic immune response and a constant susceptible cell pool (Figure 1d). In this model, we assumed free virus to be in a quasi-stationary state with respect to infected cells. This assumption allowed us to reduce the final model such that it tracks only two variables over time: CMV virus (V) and a CMV-specific immune response (E) (Equation (7)). The best version of the *VE* models 9.2 is based on Equation (9), which as with all *VE* models contains two Equations: the viral Equation features a single turnover rate $r_{v}$ and a viral killing term (meant to represent infected cell killing); the effector immune cell Equation features effector cells that proliferate in response to virus at rate $\hat{\omega }$ and die at rate $\delta_{E}$. In this version of the model, $\hat{\omega }$ is a composite parameter representing both the killing rate of infected cells and the proliferation rate of effector cells. Model 9.1 is identical to this version of the model except that ${\hat{E}}_{0}$ is estimated at zero. In model 9.2, we fixed ${\hat{E}}_{0}$ = 0, which made the model practically identifiable with all parameter RSE% < 50% (Table 1). Again, we think this value for ${\hat{E}}_{0}$ is biologically plausible based on experimental results demonstrating the absence of CMV-specific T cells at the start of viremia after HCT [18].

Model fits for the best, identifiable version of the *VE* model (version 9.2) are shown in Figure 10 using individual parameter estimates in Table A8. Population parameters estimates and RSE% are shown in Table 1. We found the population median virus turnover rate ($r_{v}$) is 0.39 day^−1^ (95% CI 0.21–0.76), equivalent to a doubling time of 1.77 days (95% CI 0.91–3.3 days). In the case of $r_{v}$, this rate should be mechanistically identical to the viral expansion slope measured in the viral kinetics section. However, the calculation of slope from data sampled relatively sparsely during the expansion phase, as in our data and in the CMV literature, is problematic [9,11,27]. Arguably, because this mathematical model contains only one parameter for viral expansion $r_{v}$, we were able to calculate the rate of expansion more reliably.

Consistent with the previous section, we found that cells in the effector compartment are long-lived with a median half-life $\left(\frac{\ln\left[2\right]}{\delta_{E}}\right)$ of 50 days.

### 3.5. CMV Turnover Rate and Immune Response Differs Relative to CMV-Related Risks Factors and Trial Outcomes

By fitting different versions of the *VE* model Equations (7)–(9), we found that this model did not outperform the *EIS*, *no TC* model in terms of AIC (Figure 11). However, we found value in this model in other dimensions. First, the parameters were mostly identifiable. Second, this model has the least variability in likelihood and AIC when running multiple assessments. Last, this model offers the benefit of a non-zero viral steady state. Particularly in the HCT setting, patients may have viral loads that initially begin to clear and later begin to rise or have persistently positive CMV viral loads [18]. This phenomenon is likely due to ongoing immunosuppression for graft-versus-host disease that some HCT recipients require long after transplant or slow engraftment of the CMV-specific T cell response from the donor [18,28]. In our data set, participant IDs 16, 31, 33, 56, 57, and 60 demonstrate this trend in the data. Because of the non-zero viral steady state, the *VE* model can capture the increasing and lingeringly positive viral loads unlike the previous models presented.

Next, because of the biological plausibility and parameter identifiability of the semi-mechanistic *VE* model, we used the best version (9.2) to ask how these CMV-specific parameters might relate to clinical risk factors for tissue-invasive CMV disease and the clinical outcome of CMV disease itself in the clinical trial. One protective factor against CMV infection and disease is a positive CMV donor serology (i.e., antibody test), meaning that the donor of the transplant has been exposed to CMV, and thus, presumably has pre-existing immunity to CMV that may confer protection to the recipient [19]. Graft-versus-host disease, a condition in which cells from the transplant attack a recipient’s skin, gastrointestinal, and liver cells, is a risk factor for CMV infection and disease. Tissue-invasive CMV disease during the first 100 days after HCT was the primary endpoint of the clinical trial.

We show the distributions of the individual parameter estimates relative to (1) CMV-donor serostatus, (2) the presence of acute graft-versus-host disease (agvh) and (3) diagnosis of CMV-disease during the clinical trial in Figure 12 and compare the mean of the individual estimates in those with and without the clinical conditions using a student’s t-test.

We found that the median CMV turnover rate ($r_{v}$) was slower (0.40 versus 0.45) when the transplant donor was CMV seropositive versus seronegative, and the proliferation rate of CMV-specific effector cells (ω) in response to virus was higher (Figure 12a), consistent with some immune protection from the CMV-positive donor.

Our model also predicts that for HCT recipients with acute graft-versus-host-disease the virus replicates faster, with doubling rates 1.2-fold higher when they have agvh (doubling times of 1.9 and 1.65 days with and without agvh, respectively) (Figure 12b), consistent with an increased risk for CMV disease in those with agvh.

Finally, we found that participants who were diagnosed with CMV disease during the trial had an 8-fold lower median immune proliferation rate in response to virus (Figure 12c).

## 4. Discussion

Despite the development of effective antiviral therapies such as ganciclovir, CMV continues to cause substantial morbidity after HCT, and viral resistance may develop over time on current therapies [2,19]. Safer, more convenient, yet potent treatments are needed. Intrahost mathematical modeling could be an important tool for understanding the dynamics of CMV virus-host interactions and has the potential to improve the clinical trials process through simulation [3,4,5,13,15]. A natural history mathematical model of CMV would allow us to perturb the model with proposed antiviral therapies and would provide a baseline for understanding required potencies and optimal dosing intervals for eliminating virus. Additionally, a baseline, quantitative understanding of the degree of immune control required for controlling virus might aid the development of a therapeutic CMV vaccine given after HCT.

Historically, mathematical modeling of CMV has been limited by the lack of availability of quantitative viral loads from untreated episodes of CMV viremia. Despite this, several groups have made important observations through modeling and other quantitative analysis of this pathogen [11,12,13,14,15,29,30,31]. However, because of this lack of data, in vivo estimation of basic model parameter values has proven difficult. In vitro estimates are unlikely to be reliable given the differences between in vitro and in vivo systems [9]. In addition, whereas viral loads from other chronic viral infections such as HIV and hepatitis C tend to both grow, plateau, and respond to antiviral therapies in stereotypic patterns, in immunocompromised hosts, such as HCT recipients, viral dynamic patterns vary with the immunologic status of the host [13,32,33].

Because we were able to obtain viral loads from frozen blood samples collected from HCT recipients from the placebo group in the historic randomized controlled trial evaluating ganciclovir for the early treatment of CMV after transplant, we have developed a natural history mathematical model for CMV after HCT [16,17]. We followed a systematic model selection procedure exploring four mechanistic ODE models with several parameterizations for a total of 38 competing models. We compared models mostly based on the Akaike Information Criteria, but in addition, we also considered the identifiability of their parameters and biological plausibility. From the competing models, we have identified two that fit the viral load data well and from which we can identify the model parameters.

In the process of fitting these models, we discovered that the data supported the inclusion of an explicit, dynamic immune system in the best models whereas a dynamic target cell compartment was not needed to recapitulate the observed data well. Rather assuming a constant, large pool of susceptible cells, whose number was unaffected by infection, was sufficient for model fitting, consistent with the fact that CMV can infect many tissue and cell types throughout the body. Given that we were fitting only to viral load data, we were limited in the number of parameters that could be identified independently. Thus, we formed some composite parameters, which limits some parameters’ independent interpretability somewhat.

For the best model with an explicit, dynamic immune system compartment and without target cell limitation (*EIS*, *no TC*, Equation (6), model 6.7), we estimated three parameters: $\hat{\beta }$, $\hat{\kappa }$, and γ. $\hat{\beta }$ is a composite parameter that describes viral infectivity, viral productivity, and the constant supply of target cells that CMV infects and thus reflects the overall viral growth rate. $\hat{\kappa }$ contains elements of both the killing rate of immune effector cells and the proliferation rate of those cells in response to virus and thus may be a marker of the adaptive immune response. Therefore, parameters $\hat{\beta }$ and $\hat{\kappa }$ may be useful in comparing participants to each other, but the actual parameter estimates may be difficult to interpret. On the other hand, γ represents a measurable rate, the clearance rate of CMV viral particles from the blood. From the best fit of this model, we estimated that the CMV genome clearance rate in plasma has a median of 0.27 per day, equivalent to a half-life of 2.6 days, which is slow. For comparison, the clearance rates of HIV and hepatitis C viruses have been estimated to be 23 per day and 8 per day, respectively [34]. Hepatitis B, a DNA virus, has complex clearance dynamics and two forms of viral DNA such that estimating the clearance rate is difficult, but the median half-life has been estimated to range from 9 to 21 h, also significantly faster than CMV [35]. The model clearance rate estimate is distinct from our viral kinetics calculation of clearance slope (equivalent to a half-life of 2.5 days) and a prior estimate made by Emery and colleagues in which the slope of viral decline in bone marrow transplant receiving ganciclovir was calculated to be equivalent to a half-life of 1.5 days. First, Emery et al. estimated this rate during treatment rather than during natural immune clearance [9]. Second, the downslope of viral load during therapy may reflect the death rate of infected cells or the removal rate of viral particles from the blood, but this cannot be disentangled without either additional knowledge of the biological system or a mechanistic model or both. Interestingly, despite this difference in calculation methods, the viral particle removal rate and the calculated viral decline kinetic from our data set in those clearing virus spontaneously was substantially slower than those receiving ganciclovir in the Emery et al. study [9]. In another study from this group, patients with HIV starting antiretroviral therapy that included a protease inhibitor and with CMV viremia were not given specific anti-CMV therapy and were followed by CMV PCR. The median time to viral clearance was 13.5 weeks (range 5–40 weeks). Granted, we cannot calculate a reliable clearance slope from this data because the median sampling interval was nine weeks, but this study further supports the slow natural clearance rate of CMV [36].

Of note, the size of the CMV DNA genome, which is considerably larger than the RNA genome of either HIV or hepatitis C, may also play a role in the slow clearance of CMV. In addition, whereas the model has allowed us to estimate this rate, a more accurate estimation could be obtained via plasmapheresis experiments as were performed in HIV and hepatitis C [34]. Not only will this parameter value be helpful to us in modeling antiviral therapy in the future but also suggests that the ability of the immune system to clear viral DNA particles from the blood after HCT is limited. In addition, in this best-fitting version of Equation (6), the death rate of immune effector cells ($\delta_{E}$) was zero. Granted the model is fit only over a period of 100 days, so biologically, it is unrealistic to conclude that these cells are immortal. However, this finding suggests that the CMV-specific cells are long-lived and may represent memory cells. The literature supports this notion with reports that CD4 and CD8 T cells are likely the most important cells for controlling CMV infection after transplant [2,18,19,21,22].

Our preferred model, the best-fitting version of the semi-mechanistic model that tracks only CMV viral load and CMV-specific effector cells (*VE* model 9.2), recapitulates the data well and contains only identifiable parameters. However, in terms of AIC, the *EIS*, *no TC* model performs better than the *VE* model. Whereas we can learn from both models, we chose to validate the *VE* model against the clinical trial risk factors and outcomes because it captures not only the viral episodes that end in complete viral clearance but also the episodes that plateau or increase after initial decrease. Additionally, the model appeared to be more stable and less sensitive to initial parameter conditions with low variability between the five assessments of each version of the model.

From the *VE* model, we estimated the viral turnover rate ($r_{v}$) as 0.39 per day, equivalent to a doubling time of 1.8 days. For HIV, the doubling time has been estimated at 1.1 days [37]. In this regard, CMV appears to replicate more slowly than HIV but surprisingly quickly for a virus usually considered to be slow in vitro [9]. Emery and colleagues found a similar median viral doubling time of 1.3 days in bone marrow transplant recipients albeit with a direct approach rather than with a mechanistic model [9]. In our viral kinetics calculations, we found the median doubling times to be 1.4 and 1.8 days depending on the calculation method (from start of estimated detectable DNAemia to first measured viral load versus to peak). However, the calculation of slope from data sampled relatively sparsely during the expansion phase, as in our data and in the CMV literature, is problematic [9,11,27]. Because this mathematical model contains only one parameter for viral expansion, $r_{v}$, this rate should estimate the true viral expansion rate. Arguably, because the model interpolates between measured points accurately, we were able to calculate the rate of expansion more reliably with this version of the model. Additionally, consistent with the *EIS*, *no TC* model, we found that immune effector cells were long-lived with a median half-life of 50 days.

We validated the *VE* model against clinical data from the randomized trial and found that both the viral turnover rate and effector cell proliferation rate in response to virus correlated with important clinical features. Those with CMV-naïve HCT donors and acute graft-versus-host disease had higher viral turnover rate parameters ($r_{v}$). Those who were diagnosed with tissue-invasive CMV disease during the clinical trial had slower proliferation of effector cells in response to virus (ω), suggesting that our model parameters have some clinical relevance.

Our study presents two intrahost models fit to long, untreated CMV reactivation episodes following HCT measured by quantitative CMV DNA PCR. In addition, we have expanded quantitative knowledge of intrahost virus-host interactions. However, there are some important limitations to this work. First, we have direct measurements only from the viral compartment and thus are limited in the number of parameters that we can identify independently. Next, our measurements from the viral compartment are measures of DNA rather than infectious virus. This limitation plagues our field, as we do not generally use viral culture clinically in humans due to problems with speed, reliability, quantification, and sensitivity. Especially in the setting of the SARS-CoV-2 pandemic, the persistence of viral genome (RNA) shedding in the absence of infectious virus has become evident [38]. Addressing this limitation is a challenge for the field of intrahost modeling.

Moving forward, we can use our best, data-validated models to simulate the ranges of viral dynamics that we observed in the placebo group while modeling the effect of ganciclovir therapy in the ganciclovir arm of the randomized trial. This will allow us to estimate the efficacy of ganciclovir and propose optimal dosing strategies for ganciclovir and other CMV antivirals. Prior to this study, we would have been unable to distinguish natural immunity from the antiviral effect.

## 5. Conclusions

We fit multiple competing versions of four ordinary differential Equation models and found two best models that recapitulated the highly variable CMV infection dynamics that occurred after HCT in the placebo group of a historic randomized trial. As a result of the fitting process, we discovered that to model this data most parsimoniously (1) an explicit, dynamic immune cell compartment was needed; (2) a dynamic target cell compartment was not needed; and (3) immune cells were long-lived. In addition, we found that viral clearance appears to be extremely slow and suggests severe impairment of the immune response after HCT. Parameters from our best model correlated well with participants’ clinical data from the trial, further validating our model.

## Figures and Tables

**Figure 1 viruses-13-02292-f001:**
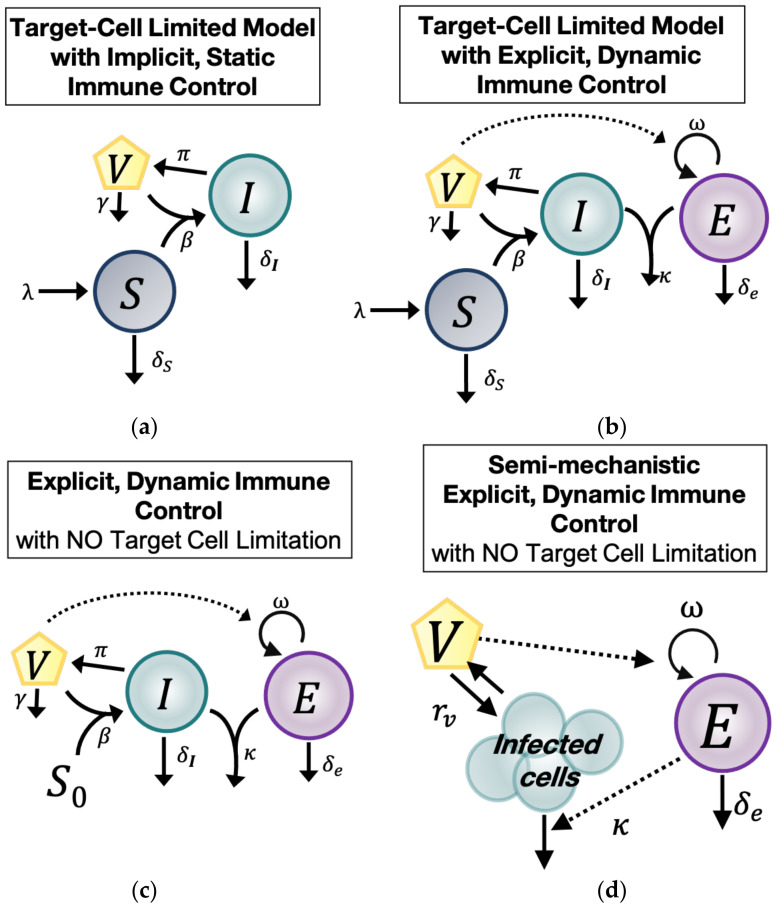
Model schematics for the main, competing structural mathematical models. (**a**) Model with target cell limitation and implicit, static immune control (*TC*, *no EIS*). (**b**) Model with target cell limitation and an explicit, dynamic immune system (*TC*, *EIS*) (**c**) Explicit, dynamic immune control model without target cell limitation (*EIS*, *no TC*). (**d**) Semi-mechanistic, explicit immune control model (*VE*).

**Figure 2 viruses-13-02292-f002:**
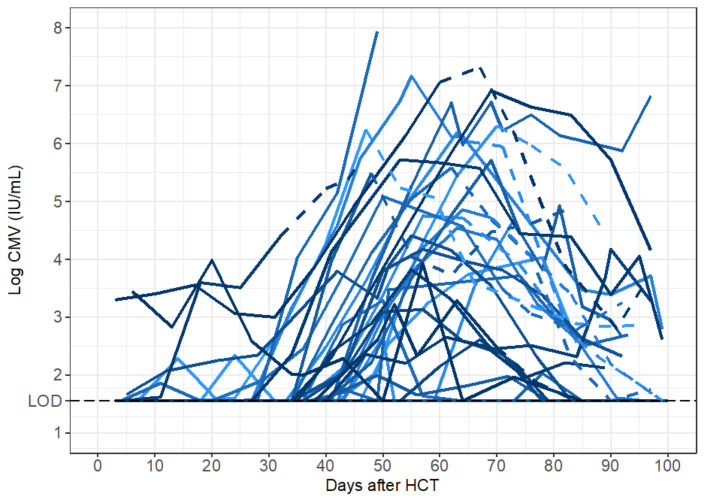
CMV viral load data from HCT recipients following transplant. Viral loads measured when no antiviral therapy was given are indicated by solid lines. Viral loads measured during or after ganciclovir was given (because participants were diagnosed with CMV disease) are indicated by dashed lines. Different shades of blue represent different participants in the trial.

**Figure 3 viruses-13-02292-f003:**
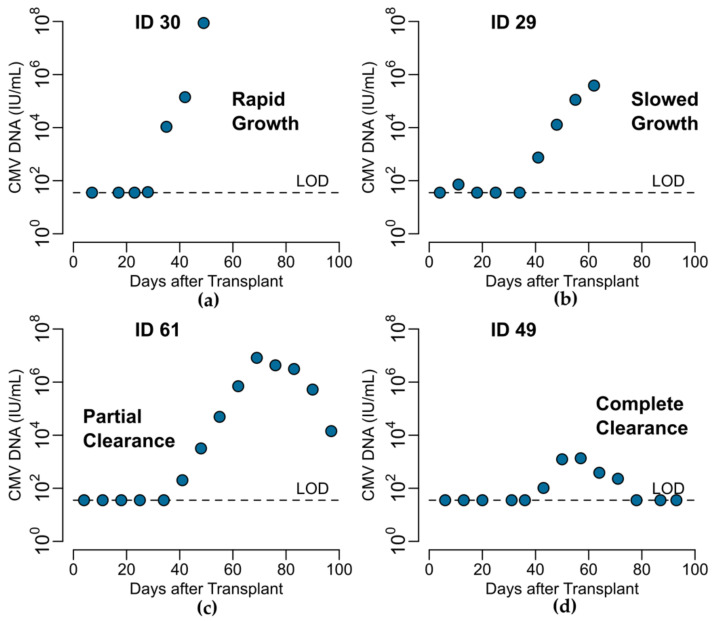
Illustrative examples of four CMV kinetics patterns in HCT recipients in the placebo group: (**a**) rapid and (**b**) slowed growth without clearance and growth followed by (**c**) partial or (**d**) complete clearance.

**Figure 4 viruses-13-02292-f004:**
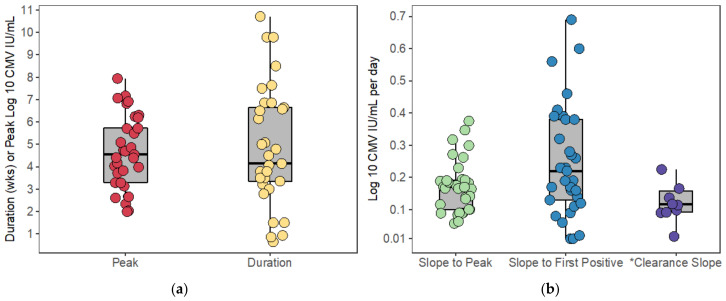
Viral kinetics of the modeled episodes. Box plots of (**a**) log 10-converted peak CMV viral load and duration of viremia in weeks load (**b**) CMV viral load slopes calculated on a log-10 converted y-axis are shown. * Clearance slopes are negative but have been multiplied by negative one here so that values can be seen on the same axis. One large outlier has been removed from each of the slope-to-first-positive and clearance-slope box plots so that the remaining values can be seen clearly. See text for maximum values. Box plots represent the distribution of the kinetics (horizontal middle line is the median; ends of the boxes represent the first and third quartiles; whiskers represent 1.5 times the interquartile range).

**Figure 5 viruses-13-02292-f005:**
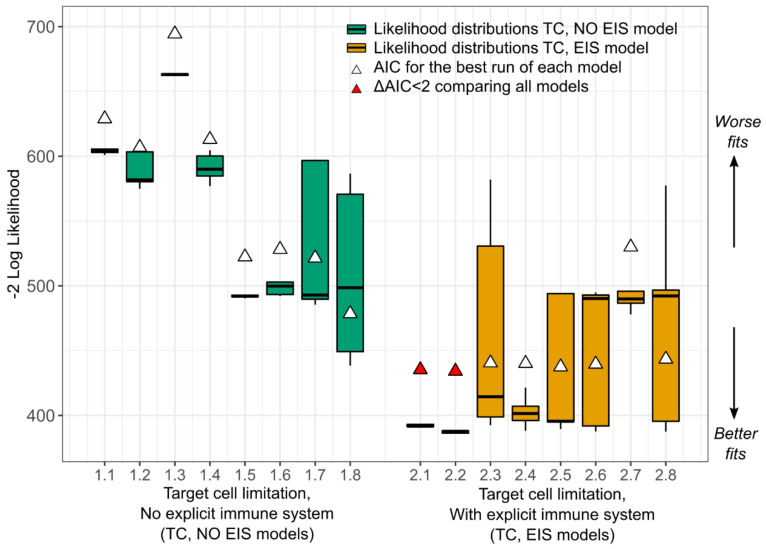
Comparison of competing models. Box plots represent the distribution of the −2logL from five fitting runs (horizontal middle line is the median; ends of the boxes represent the first and third quartiles; whiskers represent 1.5 times the interquartile range). White triangles represent the AIC with the highest logL from the five runs of each model as defined in the methods. Red triangles represent the best models by AIC defined as ΔAIC<2  relative to the model with best AIC. *TC*, *No EIS* models (Equation (1), models 1.1–1.8) are shown in green with varying parameter assumptions (Table A1). *TC*, *EIS* models (Equation (2), models 2.1–2.8) are shown in orange with varying parameter assumptions (Table A2). *EIS*: explicit immune system; *TC*: dynamic, target cell compartment.

**Figure 6 viruses-13-02292-f006:**
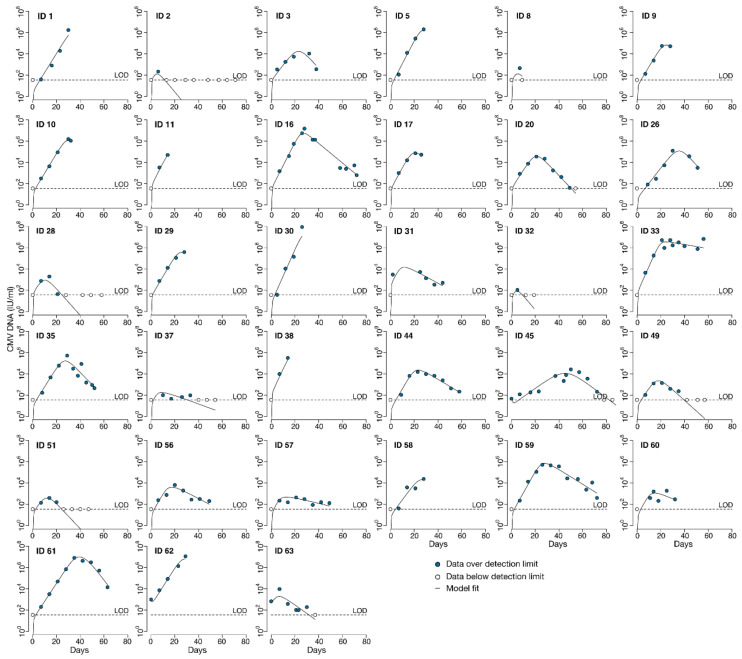
Fit of the best *TC*, *EIS* model (2.2) to CMV DNA viral loads from untreated recipients of HCT following transplant. Solid lines represent model predictions; blue circles represent observed viral loads above the assay limit of detection (LOD); white circles indicate observed viral loads below the LOD.

**Figure 7 viruses-13-02292-f007:**
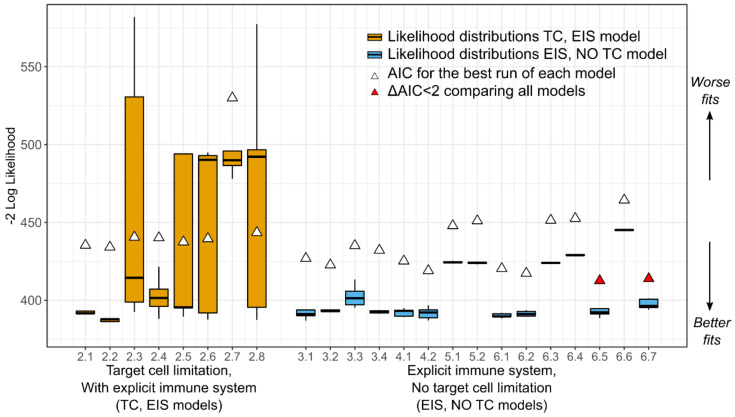
Comparison of competing models. Box plots represent the distribution of the −2logL from five fitting runs (horizontal middle line is the median; ends of the boxes represent the first and third quartiles; whiskers represent 1.5 times the interquartile range). White triangles represent the AIC with the highest logL from the five runs of each model as defined in the methods. Red triangles represent the best models by AIC defined as ΔAIC<2  relative to the model with best AIC. The target cell-limited models (*TC*, *EIS*) are shown in orange as in Figure 1b. The models with constant susceptible cells (*EIS*, *No TC*—Equations (3)–(6); models 3.1–3.4, 4.1–4.2, 5.1–5.2, 6.1–6.7) are shown in blue with varying parameter assumptions (Table A2). *EIS*: explicit immune system; *TC*: dynamic target cell compartment.

**Figure 8 viruses-13-02292-f008:**
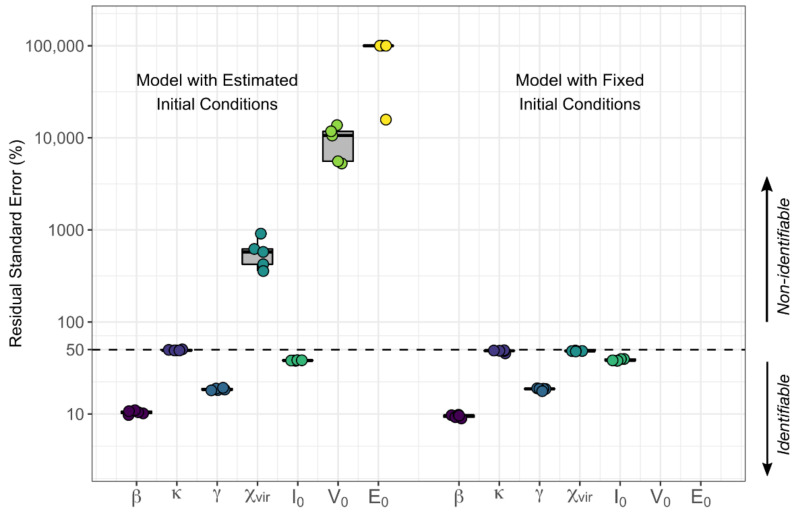
Distributions (over five assessments) of relative standard error percentages (RSE%) of estimated parameters in the *EIS*, *no TC* model version 6.2 with estimated initial conditions for state variables on the left and model version 6.7 with initial values for $\hat{E}$ and V fixed at zero on the right. Generally, parameters with RSE% above 50% (dashed line) are considered non-identifiable. Note that RSE% are not shown for fixed ${\hat{E}}_{0}$ and $V_{0}$ on the right because they were fixed and not estimated.

**Figure 9 viruses-13-02292-f009:**
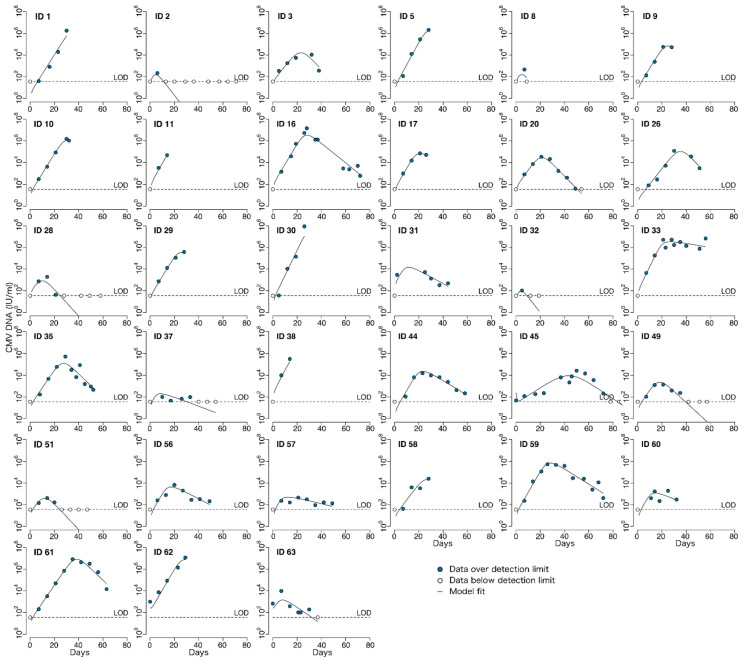
*EIS*, *no TC* model fits from model 6.7 to CMV DNA viral loads from untreated recipients of HCT following transplant. Solid lines represent model predictions; blue circles represent observed viral loads above the assay limit of detection (LOD); white circles indicate observed viral loads below the LOD.

**Figure 10 viruses-13-02292-f010:**
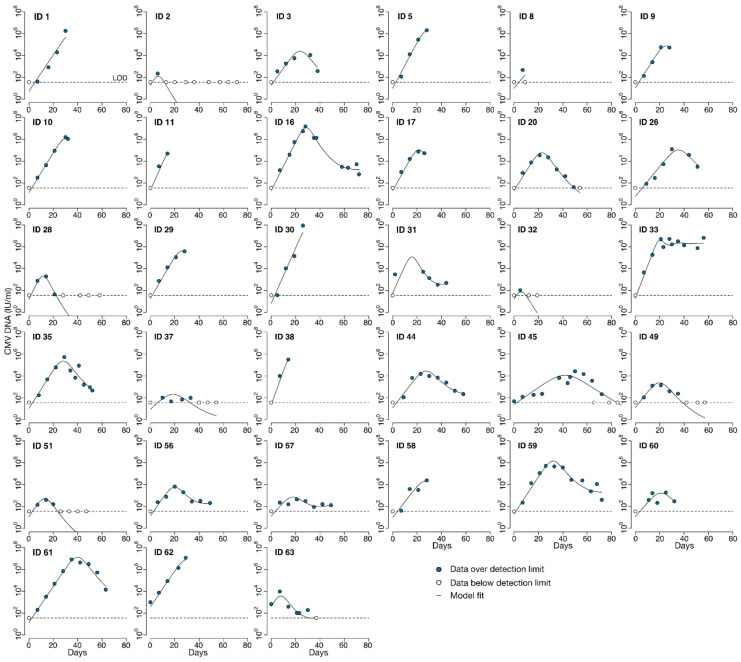
*VE* model (9.2) fits to CMV DNA viral loads from untreated recipients of HCT following transplant. Solid lines represent model predictions; blue circles represent observed viral loads above the assay limit of detection (LOD); white circles indicate observed viral loads below the LOD.

**Figure 11 viruses-13-02292-f011:**
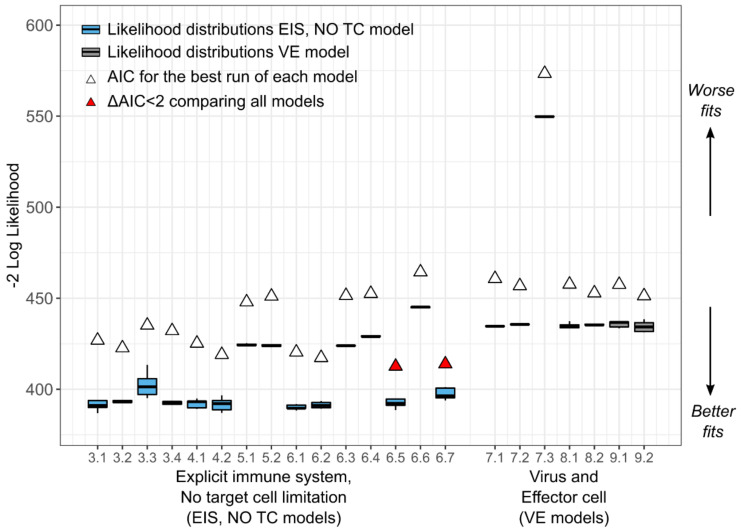
Comparison of competing models. Box plots represent the distribution of the −2logL from five fitting runs (horizontal middle line is the median; ends of the boxes represent the first and third quartiles; whiskers represent 1.5 times the interquartile range). White triangles represent the AIC with the highest logL from the five runs of each model as defined in the methods. Red triangles represent the best models by AIC defined as ΔAIC<2  relative to the model with best AIC. The mechanistic models with dynamic immunity and no target cell limitation (*EIS*, *No TC*) are shown in blue as in Figure 1c. The *VE* models are shown in grey (Equation (7), models 7.1–7.3, 8.1–8.2, 9.1–9.2). *EIS*: explicit immune system; *TC*: dynamic target cell compartment; *VE*: virus and effector cell.

**Figure 12 viruses-13-02292-f012:**
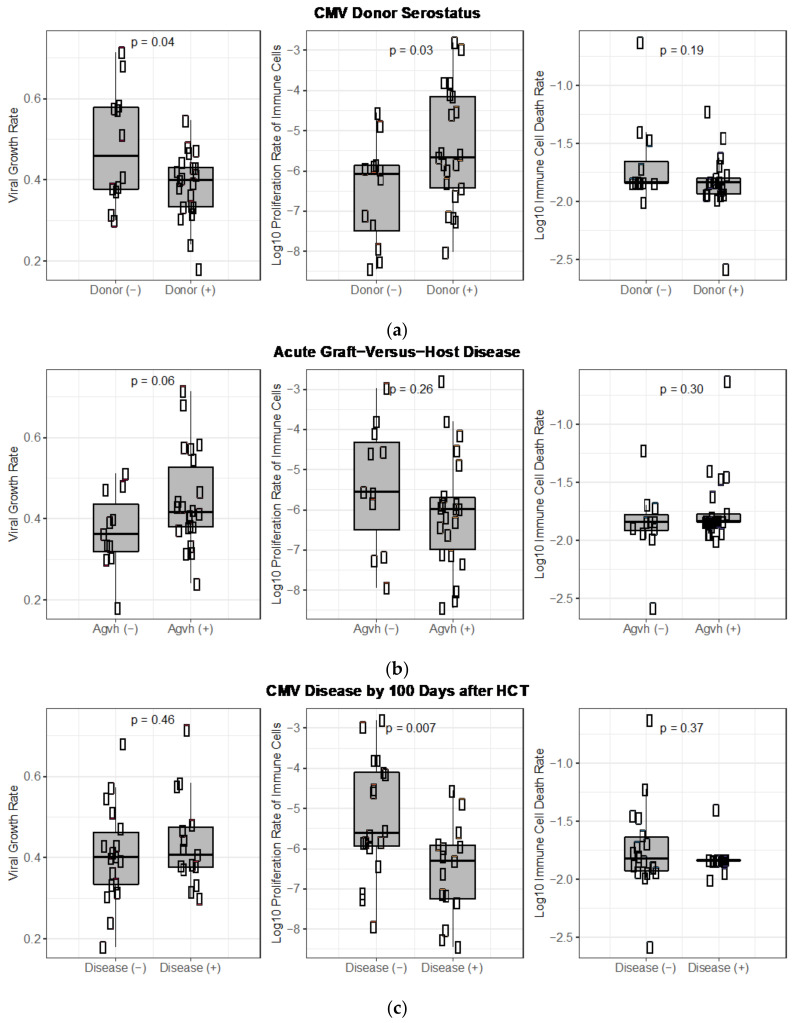
Model parameter estimates (viral growth rate, CMV-specific effector cell proliferation rate, and death rate of effector cells) compared among individuals relative to clinical CMV risk factors and outcomes: (**a**) CMV donor serostatus, (**b**) acute graft-versus-host disease and (**c**) CMV disease diagnosed during the randomized trial.

**Table 1 viruses-13-02292-t001:** Population parameter estimates of the best version of the *VE* model 9.2 using Equation (9). RSE%: percentage of the relative standard error. An RSE% > 50 implies the corresponding parameter might not be identifiable with the available data.

Parameter		Value	RSE%
Fixed effects $\left(\theta^{pop}\right)$	$r_{v}^{pop}$	0.39	6.7
	$\omega^{pop}$	1.1 × 10^−6^	40.9
	$\delta_{E}^{pop}$	0.014	39.4
	$V_{0}^{pop}$	14.7	23.5
Covariate	$\chi_{viremic}$	2.3	28.1
Standard deviation of the random effects $\left(\sigma_{\theta }\right)$	$\sigma_{r_{v}}$	0.33	16.2
	$\sigma_{\omega }$	4.0	15.8
	$\sigma_{\delta_{E}}$	1.3	25.4
	$\sigma_{V_{0}}$	0.88	44.0
CMV viral load measurement error	$\sigma_{v}$	0.43	4.1

## Data Availability

The data presented in this study are available on request from the corresponding author.

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
