# Peer review of "Mathematical Modeling of Within-Host, Untreated, Cytomegalovirus Infection Dynamics after Allogeneic Transplantation"

_viruses, 2021, doi:10.3390/v13112292_

Round 1

Reviewer 1 Report

The revised manuscript has been improved. 

A few comments: 

lines 643 and 645 : please check whether genome clearance rate unit should be per day (i.e., without "virions").  

line 644: not clear the note "...which is slow" . Why it is compared to HIV and HCV clearance (RNA viruses) and not DNA viruses such as HBV  ? 

Author Response

Reviewer 1

The revised manuscript has been improved. 

We appreciate the reviewer taking time to review our manuscript again. The initial review helped us strengthen the paper greatly, and the second review has led us to make further improvements. Thank you!

A few comments: 

lines 643 and 645 : please check whether genome clearance rate unit should be per day (i.e., without "virions").  

We thank the reviewer for noticing this mistake. We have corrected it in the revised manuscript.

line 644: not clear the note "...which is slow" . Why it is compared to HIV and HCV clearance (RNA viruses) and not DNA viruses such as HBV  ? 

We have now included estimates of the half-life of hepatitis B virus to give a comparison relative to a DNA virus. We thank the reviewer for pointing this out. We are most familiar with the HIV and HCV literature, which is why we made those more familiar comparisons initially.

Reviewer 2 Report

Review of the paper "Mathematical modeling of within-host, untreated, cytomegalovirus infection dynamics after allogeneic transplantation" co-authored by E.R. Duke, F.A.T. Boshier, M. Boeckh, J.T. Schiffer and E. F. Cardozo-Ojeda.

In this paper, the authors propose a list of models based upon differential equations to understand the dynamics and the natural history of the cytomegalovirus in patients with marrow transplantation. Then, using data from placebo patients they perform model calibrations using the model selection package Monolix. Finally, they disscus and conclude which model terms describe the cytomegalovirus dynamics.

This paper has some issues to be addressed. 

* The points 2.4 - 2.7 about modeling and simplification of the model are confusing. The authors propose two models and the second is simplified in several ways assuming that some model parameters are zero. It is unclear if all the models are used and when, until the Appendix. It is unclear if the assumptions about that some model parameters are zero are biologically meaningful.

* The authors claim that they are going to compare 38 models (in fact, variations of the same), but the only way to check all of these models are in three tables located in the Appendix. It does not seem to help the reading.

* All the models considered are simplifications/manipulations of the model (2). Maybe, it would be better to consider few models and include scenarios where the model parameter values take particular values.

* Initial conditions in ODEs are very sensitive. Small variations may lead to large outcomes, in particular, in the linear equations where the growth is exponential. Hence, an explanation why the authors consider only the values appearing in the tables in the Appendix would be welcome.

* The model selection is an open problem. It seems that the "philosophy" behind the approach used is to find the best fitting as long as the model has biological and well supported meaning. This is a good "philosophy". Also, the model fittings obtained are fairly well. However, it seems that the "errors" (logarithm of the likelihood) obtained in the fitting are a bit high. If possible, a comment about that would be welcome. 

If some of the above issues have been misunderstood, sorry. But this reviewer has found the reading of the paper difficult and, therefore, it would be interesting to consider a clearer structure of the paper where the goals are established at the beginning and followed along the paper. 

Author Response

Reviewer 2

Review of the paper "Mathematical modeling of within-host, untreated, cytomegalovirus infection dynamics after allogeneic transplantation" co-authored by E.R. Duke, F.A.T. Boshier, M. Boeckh, J.T. Schiffer and E. F. Cardozo-Ojeda.

In this paper, the authors propose a list of models based upon differential equations to understand the dynamics and the natural history of the cytomegalovirus in patients with marrow transplantation. Then, using data from placebo patients they perform model calibrations using the model selection package Monolix. Finally, they disscus and conclude which model terms describe the cytomegalovirus dynamics.

We appreciate the reviewer’s time and consideration. We have addressed the reviewer’s concerns below and have improved the paper in response. We apologize that our paper was difficult to read and hope it is now clearer.

This paper has some issues to be addressed. 

* The points 2.4 - 2.7 about modeling and simplification of the model are confusing. The authors propose two models and the second is simplified in several ways assuming that some model parameters are zero. It is unclear if all the models are used and when, until the Appendix. It is unclear if the assumptions about that some model parameters are zero are biologically meaningful.

We apologize that the methods section did not draw enough of a distinction between the four main models to make our approach clear. We have edited the introductory sentences of each main model section to point out the differences between the main models to clarify this. Each of the main models are biologically and mathematically distinct. The second model includes an additional compartment from the first. With certain assumptions, the third and fourth model can be derived from the second. However, these assumptions are biologically and mathematically important. Within each main model, we set select parameters to zero and fit each of those instances of the model to determine which instance of the main model the data support most strongly (Methods section 2.9 “Model Fitting”). We fit all 38 versions of the models and display those results in the Appendix and in the main manuscript figures (Figures 5, 7, and 10).

In the model selection process, within each main model, we chose which parameters could plausibly be zero in a biological setting and set those to zero individually and in combination such that we fit every combination of biologically feasible parameters to determine what version of the model the data supported most strongly. In response to the reviewer’s comment, we have included the previous sentence in the methods (section 2.1) and have explicitly described which parameters are fixed to 0 in combination with other parameters in the model selection process and the biological significance of this in section 2.9 of the methods “Model fitting.” We think the methods are more explicit and clearer now and appreciate this comment.

In addition, where relevant, we discuss the significance of parameters being equal to zero in the results for the two, best-fitting, identifiable models.

* The authors claim that they are going to compare 38 models (in fact, variations of the same), but the only way to check all of these models are in three tables located in the Appendix. It does not seem to help the reading.

We felt that including the tables of all model fits in the main paper would be overwhelming for readers but made these results available in the Appendix. We have also displayed these results graphically by AIC and log-likelihood in the figures in the main manuscript.

* All the models considered are simplifications/manipulations of the model (2). Maybe, it would be better to consider few models and include scenarios where the model parameter values take particular values.

We apologize that this was not clear. However, each of the four main models is mathematically and biologically distinct. The main point of the paper is to show the difference in fits between models that include target-cell limitation as the only means for controlling viremia versus models that include a dynamic effector cell immune response as the only means for controlling viremia versus models that include both. The final model includes only a dynamic effector cell immune system but lacks an explicit infected cell compartment. The differences between the main models are shown visually in Figure 1. Important assumptions must be made to transition between those models. These assumptions are not trivial biologically or mathematically. We hope we have made this approach clearer with language changes in the methods.

In response to the second point, we considered as many models as were biologically plausible because of unique richness of this data set (Methods 2.2 “Clinical Data”) and the opportunity it afforded us in fitting biological parameters that have not been able to be identified previously. Thus, we wanted to be thorough and not make assumptions about either model structure or parameter values since prior values have been based on data that was not as rich or on in vitro data, which is known to differ greatly from in vivo data.

* Initial conditions in ODEs are very sensitive. Small variations may lead to large outcomes, in particular, in the linear equations where the growth is exponential. Hence, an explanation why the authors consider only the values appearing in the tables in the Appendix would be welcome.

Initial conditions were estimated for all state variables. We have added a phrase indicating this specifically in the methods, section 2.9. In the tables in the appendix, the initial conditions shown are estimated from the fitting procedures and were not fixed except as indicated in the table legends and discussed in the text for the two best models.

* The model selection is an open problem. It seems that the "philosophy" behind the approach used is to find the best fitting as long as the model has biological and well supported meaning. This is a good "philosophy". Also, the model fittings obtained are fairly well. However, it seems that the "errors" (logarithm of the likelihood) obtained in the fitting are a bit high. If possible, a comment about that would be welcome. 

Generally, the calculation of likelihood depends on many elements: the model structure, the number of parameters in the models, and the number of data points in the whole population (longitudinal time points over all individuals measured), the nature of the data points (censored values below the limit of detection increase uncertainty), and the methods used by the fitting software (in this case, Monolix). Thus, it is impossible to declare the goodness of fit of a model based on likelihood alone. Rather, likelihoods are used to calculate AIC that are then used to compare models that have been fit on the same data set with the same fitting algorithms. As the reviewer points out, this is the philosophy of model selection that we have used, comparing AIC of the models to choose the best fitting and most biologically plausible. In addition, we also considered model identifiability as a third criterion.

If some of the above issues have been misunderstood, sorry. But this reviewer has found the reading of the paper difficult and, therefore, it would be interesting to consider a clearer structure of the paper where the goals are established at the beginning and followed along the paper. 

In Methods section 2.1 “Study Approach”, we outline the structure of the paper and establish our goal to characterize the natural history of CMV in untreated bone marrow transplant recipients. We use four main ODE models and fit biologically plausible versions of those models. Then, we use model selection theory to determine the best fitting, identifiable models. We have clarified the language in the methods and hope that the reviewer finds the paper more enjoyable to read going forward.

This manuscript is a resubmission of an earlier submission. The following is a list of the peer review reports and author responses from that submission.

Round 1

Reviewer 1 Report

This is an elegantly conducted modeling study, supported by new measurement of historic randomized controlled trial samples. Duke et al. used DNA PCR to measure viral load dynamics from samples of untreated cytomegalovirus (CMV) infection in recipients of allogeneic hematopoietic cell transplantation (HCT). The data was fit to 38 competing models, and the best models were chosen according to both model selection theory and knowledge of CMV. The authors emphasize the slow clearance of CMV (in HCT recipients without antiviral treatment) as a new finding, and point out several other findings (persistence of immune effector cells; surprisingly fast viral turnover in vivo versus in vitro) consistent with the literature. To support their selected model, Duke et al correlate model parameters and patient clinical data such as CMV donor serostatus and disease outcomes. I have two major comments: one regarding the new finding, and the other regarding proposed extensions of the work.

Major comments:

  1. This study uses CMV genomes (assayed by PCR amplification of CMV DNA in frozen serum samples) as a proxy for infectious CMV virions. Although PCR is valued for diagnosis of CMV (Miller JM et al. Clinical Infectious Diseases. 2018. PMID: 29955859) particularly due to its speed and sensitivity relative to viral culture (Fajac A et al. Bone Marrow Transplantation. 1997, PMID: 9337060; de Vries JJC et al. J Clin Virol. 2012, PMID: 22177273; Ross SA et al. J Infectious Disease. 2014. PMID: 24799600), the presence of viral genomes does not necessarily correspond to the presence of infectious virus, which could be detected by e.g. plaque-forming assays or viral culture. As an example, SARS-CoV-2 RNA shedding has persisted long after replication-competent virus is cleared (Owusu D et al. J Infectious Disease. 2021. PMID: 33649773). Given the emphasis on slow immune clearance of CMV as a new finding (lines 548-551), the authors need to provide evidence that slow clearance of CMV genomes corresponds to slow clearance of replication-competent CMV virions.
  2. The study provides a 'baseline' mathematical model of infection by focusing on untreated CMV viremia post HCT. The authors state that they could use their best models to estimate the efficacy of antiviral therapies in clinical trials, and describe plans to apply this to ganciclovir data. But it seems that this model selection approach should be applied once more to data in the untreated and treated arms. This would (1) ensure that no idiosyncrasies of the clinical dataset result in a more optimal model, and (2) assess whether all the assumptions of the 'baseline' model hold true when introducing ganciclovir treatment.

Minor comments:

Line 75: typo “ordinarily” to “ordinary”

Line 154-155: typo “equation 6” to “equation 7”

Line 163-164: Why are these parameters modeled by multiplicative scaling of exponentiated noise? Is it so that the noise becomes additive when the viral load is log10-transformed, or is there another reason?

Line 167: What does it mean to estimate χviremic as a covariate of V0?

Line 192-193: It may help readers to provide a short description of how to interpret the residual squared error in the context of practical identifiability. The actual calculation is somewhat opaque (i.e. Monolix seems to use the square root of the diagonal elements of the inverse of the Fisher Information Matrix to compute standard errors for each parameter, where the FIM is estimated via the SAEM approach (Stochastic Approximation of the Expectation Maximization).

Confusing logic at line 334: “…arrive at the model with the best AIC, model 6.2” This statement is a little confusing since the reader can see that models 6.5 and 6.7 had better AIC versus model 6.2 in both Figure 5 and Table A5. The subsequent explanations make it sound like models 6.3-6.7, with fixed initial conditions, were only explored afterwards. 

Line 357: “Because we included the covariate χviremic, all model V0 values fixed at zero are fit to observed values below the limit of detection.” What does this mean?

Reviewer 2 Report

Elizabeth Duke and colleagues mathematically modeled CMV DNA kinetics during placebo treatment from a historic trial by Goodrich et al (NEJM 1991), using frozen samples that were recently PCR quantified from frozen samples and kinetically analyzed by the same group (Duke et al. J Clin Invest 2021). The authors developed several mathematical models and calibrated them with the newly measured viral data.  They used model selection theory to predict which model is the most plausible one to describe the viral-host dynamics. The availability of longitudinal CMV DNA kinetics from more than 30 patients without antiviral treatment for ~100 days (compared to Emery et al. Lancet 2000 historical kinetic analysis of ~25 days) appears novel. Thus, the current modeling approach to explain the viral dynamics also appears the first modeling attempt in the field which is exciting. I have several comments.

  1. The description of viral kinetics in section 3.1 is not well described and seems not complete (e.g., what about patients with viral increase followed by a plateau and those with partial viral clearance that reach a new lower plateau). There are no details provided about the viral kinetic characterization (slopes and peak levels etc) among the identified kinetic groups proposed in Fig. 2. Also, why data were available/presented only during viral increase (e.g., Fig. 2, patients 1 and 29 ..and others shown in the fig. 4)?
  2. The authors have recently published a detailed kinetic analysis (Duke et al. J Clin Invest 2021) in which the kinetics were also associated with disease outcomes but surprisingly these seem not to be presented or acknowledged in the current manuscript. I suggest including a section regarding disease outcomes in the placebo group and their association with viral kinetic patterns.
  3. The math models were described both in the Methods and the Results sections. I suggest reducing redundancy in the manuscript (e.g., present model equations and their schematic diagrams together). I believe the “four models” that the authors refer to are Eqs. 1, 2, 3 and 9 but this needs to be clearly described.
  4. Estimated model parameters are not described in the main text nor compared/explained. For example, why pi is order of magnitude higher in Table A4 compared to Table A3. Similar question about V0.
  5. Model predictions of S, I and/or E are not shown along with V for each best model fit. These are important aspects to be presented and discussed, especially the biological/immunological predictions and their associations with clinical outcomes.
  6. The author stated that the validated the model with disease outcomes (Fig. 10). This is not clear since the models do not include any clinical aspects that can be validated. In fact, since the authors have also recently measured and analyzed the antiviral treatment arm of the historical trial (Duke et al. J Clin Invest 2021) they may want to consider validating their models (at least the best model) with the CMV kinetics under antiviral treatment.
  7. The Discussion section needs to be revised as it reads like a repetition of the previous sections. Also, the authors seem to compare their parameter estimations (e.g., viral clearance rate) with HIV and HCV viruses rather with previous studies on CMV. It seems that the current predictions of the model regarding viral doubling time and clearance rate are within the range of the findings made by Emery et al. Lancet 2000 (ref 11 in the current manuscript). Lastly, it would be important to focus more in the Discussion on the insights that the modeling approach provide beyond the Lancet 2000 kinetic findings and other relevant studied in the field.